# Alignment-Free Method to Predict Enzyme Classes and Subclasses

**DOI:** 10.3390/ijms20215389

**Published:** 2019-10-29

**Authors:** Riccardo Concu, M. Natália D. S. Cordeiro

**Affiliations:** LAQV@REQUIMTE/Department of Chemistry and Biochemistry, Faculty of Sciences, University of Porto, 4169-007 Porto, Portugal

**Keywords:** QSAR, machine learning, artificial neural network, enzyme, enzyme classification, alignment-free

## Abstract

The Enzyme Classification (EC) number is a numerical classification scheme for enzymes, established using the chemical reactions they catalyze. This classification is based on the recommendation of the Nomenclature Committee of the International Union of Biochemistry and Molecular Biology. Six enzyme classes were recognised in the first Enzyme Classification and Nomenclature List, reported by the International Union of Biochemistry in 1961. However, a new enzyme group was recently added as the six existing EC classes could not describe enzymes involved in the movement of ions or molecules across membranes. Such enzymes are now classified in the new EC class of translocases (EC 7). Several computational methods have been developed in order to predict the EC number. However, due to this new change, all such methods are now outdated and need updating. In this work, we developed a new multi-task quantitative structure–activity relationship (QSAR) method aimed at predicting all 7 EC classes and subclasses. In so doing, we developed an alignment-free model based on artificial neural networks that proved to be very successful.

## 1. Introduction

By the late 1950s, the International Union of Biochemistry and Molecular Biology foresaw the need for unique nomenclature for enzymes. In those years, the number of known enzymes had grown very rapidly and, because of the absence of general guidelines, the nomenclature of the enzymes was getting out of hand. In some cases, enzymes with similar names were catalyzing different reactions, while conversely different names were given to the same or similar enzymes. Due to this, during the third International Congress of Biochemistry in Brussels in August 1955, the General Assembly of the International Union of Biochemistry (IUB) decided to establish an International Commission in charge of developing a nomenclature for enzymes. In 1961, the IUB finally released the first version of the Enzyme Classification (EC) and Nomenclature List. This nomenclature was based on assigning a four number code to enzymes with the following meaning: (i) the first number identifies the main enzyme class; (ii) the second digit indicates the subclass; (iii) the third number denotes the sub-subclass; and (iv) the fourth digit is the serial number of the enzyme in its sub-subclass. Six enzyme classes were identified, with the classification based on the type of reaction catalyzed: oxidoreductases (EC 1), transferases (EC 2), hydrolases (EC 3), lyases (EC 4), isomerases (EC 5) and ligases (EC 6) [1]. Although several revisions have been made to the 1961 version, the six classes identified have not received any change. However, in August 2018, a new class was added. This new class contains the translocases (EC 7), and was added to describe those enzymes catalyzing the movement of ions or molecules across membranes or their separation within membranes. For this reason, some enzymes which had previously been classified in other classes—EC 3.6.3 for example—were now included in the EC 7 class. 

Predicting enzyme classes or protein function using bioinformatic tools is still a key goal in bioinformatics and computational biology due to both the prohibitive costs and the time-consuming nature of wet-lab-based functional identification procedures. In point of fact, there are more than four thousand sequences whose function remains unknown so far and this number is still growing [2]. The problem is that our ability to assign a specific function to a sequence is far lower than our ability to isolate and identify sequences. For this reason, significant efforts have been devoted to developing reliable methods able to predict protein function. 

Several methodological strategies and tools have been proposed to classify enzymes based on different approaches [3,4,5,6,7,8,9,10]. The Basic Local Alignment Search Tool (BLAST) [11] is likely to be one of the most powerful and used tools which finds regions of similarity between biological sequences. The program compares nucleotide or protein sequences to sequence databases and calculates their statistical significance. However, as is the case with all methods, these procedures may fail under certain conditions. In some cases, enzymes with a sequence similarity higher than 90% may belong to different enzyme families and, thus, have different EC annotations [12,13,14]. On the other hand, some enzymes which share the same first EC number may have a sequence similarity below 30%. Some authors have described this situation well and highlighted the need to develop alignment-free methods, which may be used in a complementary way [15,16]. Other relevant tools based on sequence similarity are the UniProtKB database [17], the Kyoto Encyclopedia of Genes and Genomes (KEGG) [18], the PEDANT protein database [19], DEEPre [20], ECPred [21] and EzyPred [22]. DEEPre is a three-level EC number predictor, which predicts whether an input protein sequence is an enzyme, and its main class and subclass if it is. This method is based on a dataset of 22,198 sequences achieving an overall accuracy of more than 90%. ECPred is another enzymatic function prediction tool based on an ensemble of machine learning classifiers. The creators of this tool developed it using a dataset of approximately 245,000 proteins, achieving score classifications in the 6 EC classes and subclasses like the ones reported by DEEPre. EzyPred is a top-down approach for predicting enzyme classes and subclasses. This model was developed using a 3-layer predictor using the ENZYME [23] dataset (approximately 9800 enzymes when the model was developed), which was able to achieve an overall accuracy above 86%. Other relevant methods with similar classification scores have also been reported [10,15,20,24,25]. All these methods have proved to be robust; however, they are all outdated since they cannot predict the EC 7 classification, and should therefore be updated in accordance with the new EC class.

In light of what has been referred to so far, the major target of this work was to develop an alignment-free strategy using machine learning (ML) methods to predict the first two digits of the seven EC classes. Previous ML methods have used alignment-free numerical parameters to quantify information about the 2D or 3D structure of proteins [26,27,28,29]. Specifically, Graham, Bonchev, Marrero-Ponce, and others [30,31,32,33,34] used Shannon’s entropy measures to quantify relevant structural information about molecular systems. In addition, González-Díaz et al. [35,36,37] introduced so-called Markov–Shannon entropies (*θ_k_*) to codify the structural information of large bio-molecules and complex bio-systems or networks. For comparative purposes, we developed different linear and non-linear models, including a linear discriminant analysis (LDA) and various types of artificial neural networks (ANNs). In addition, we focused our work on performing an efficient feature selection (FS). Nowadays, there are several software packages or tools that may be used to calculate thousands of molecular descriptors (MDs). As a result, a proper FS method is essential to develop robust and reliable quantitative structure–activity relationship (QSAR) models. This is particularly the case when using ANNs, since QSAR models developed with a large set of MDs are really complex, vulnerable to overfitting and difficult to obtain a mechanistic interpretation from [38,39].

## 2. Results

### 2.1. LDA Model

As a first step, we used the LDA algorithm implemented in the software STATISTICA^®^ [40] to derive a linear model able to discriminate all of the subclasses of enzymes using a multi-task model, which means that a single model was developed in order to assign each enzyme to a specific class. From the first pool of more than 200 variables, we selected four that clearly had an influence on the model using a supervised forward stepwise analysis. In order to validate the model, we split our dataset, assigning 70% of the entries to the training class and the remaining 30% to the validation class. The latter was used for validation of the model using a cross-validation procedure. The LDA model had the following overall values for specificity: Sp = 99.71%, sensitivity: Sn = 98.16% and accuracy: Acc = 98.66%. In the training series, the model displayed Sp = 99.71%, Sn = 98.13% and Acc = 98.63%, while in the validation series Sp = 99.71, Sn = 98.27, Acc = 98.73. All of these statistics are reported in Table 1.

The linear equation (Equation (1)) for this model is shown below and information regarding its variables is given in Table 5:(1)$$EC=\;<Tr3\left(srn\right)>*-0.95+<Tr5\left(srn\right)*-0.80+DTr5\left(srn\right)*-0.80+Dtr3\left(srn\right)*1.01-2.05$$Other relevant statistics for the LDA model (both training and validation), such as the Wilk’s lambda and Matthews correlation coefficient (MCC), are reported in Table 2.

### 2.2. ANN models

We then decided to move a step forward and try to develop non-linear models using various neural networks’ architectures. We firstly investigated ANN models using either the multi-layer perceptron (MLP) algorithm or the radial basis function (RBF) [41,42,43,44,45,46]. To do so, we ran a set of 50 ANN-MLP models in order to identify the best topology and architecture. The best model found had an MLP 4-9-2 topology, and was developed using the same four variables used for the LDA model. Additionally, it was able to correctly classify 100% of the cases in both the training and validation series. Table 3 shows the statistical parameters obtained for this model. As can be seen, the MCC value was, as expected, 1.

For comparative purposes, Table 4 reports the statistics of the 10 best MLP and RBF models found.

The results reported in Table 4 clearly indicate that MLP models perform better than RBF ones. Even if the best MLP model was able to achieve 100% overall accuracy, we decided to perform a quantitative analysis to infer whether the MLP models were failing. As can be seen in Table 5, the non-optimal MLP models were particularly problematic in discriminating the EC 6.5 subclass. 

Finally, a sensitivity analysis was also performed to assess the influence of the MDs in the model. The results of this analysis are shown in Table 6. 

Sensitivity analysis refers to the assessment of the importance of predictors in a developed model, with higher values of sensitivity being assigned to the most important predictors. As seen, the high sensitivity values found for some of the parameters suggest that the model’s performance can drastically fall if the parameters used in the model are removed. On the other hand, parameters with lower values of sensitivity may be discarded since they are not relevant to the performance of the model and may lead to an overfitted model. Regarding the variables presented in Table 6, they are traces of the *n* connectivity matrices of the amino acid sequences. The terms 3 and 5 represent the order of the matrix used in the calculation. The terms within brackets (“< >”) represent the mean value of each subclass, while “D” stands for the difference (or distance) between each amino acid sequence and the mean value of its subclass. This basically means that the model, in order to correctly predict each sequence as an enzyme and then input it into the specific subclass, is calculating the distance between each input and the mean of its subclass. This is in fact how a multi-target model works.

## 3. Discussion

The main aim of this study was to develop a new QSAR-ML model able to predict enzyme subclasses considering the new and recently introduced EC class 7. We retrieved from the Protein Data Bank (PDB) more than 26,000 enzyme and 55,000 non-enzyme sequences in order to build up our dataset. All of the enzyme sequences belonged to one of the 7 main classes and 65 subclasses. The EC 7 class was introduced just few months ago and, due to this, all of the current models do not include this new enzyme class. As a result, the classification or prediction such models are performing may be misleading. Hence, the development of new models which are capable of predicting all enzyme classes and subclasses—including the EC 7 class—are of utmost importance. In view of this, we developed a new machine learning model able to discriminate between enzymes and non-enzymes. In addition, the model was capable of assigning enzymes to a specific enzyme subclass. We generated linear and non-linear models using alignment-free variables to find the best model to predict EC classes and subclasses. The results of the linear model were impressive since with only four MDs the model could discriminate between enzymes and non-enzymes, as well as assign a specific EC class and subclass to each enzyme sequence. We checked the accuracy and robustness of the model and the results clearly indicate that the model is reliable. Regarding the validation, we performed a classical cross-validation procedure using 30% of the dataset. This led to almost the same results for the training and validation sets, indicating once more the robustness of the model and approach.

Although the accuracy of the derived LDA model was near 100%, we decided to further test our approach by developing some neural network models, which usually improve LDA results. To the best of our knowledge, an MLP is generally considered the best ANN algorithm and, in this case, had the potential to improve our linear model. As previously reported, the MLP was able to perfectly discriminate between enzymes and non-enzymes, in addition to assigning each enzyme sequence to a specific subclass. It is also remarkable that the best model only needed nine neurons in the hidden layer. This low number of neurons, considering the number of sequences and variables, suggest that the model is not suffering from an overfitting problem. Mechanistic interpretation of ANN models is always a challenging task since these models do not lead to simple linear equations. A sensitivity analysis may then be used to analyze the influence of each MD on the model. For the ANN model, we carried out such an analysis to evaluate the weight of each variable in the model. This analysis is also useful for identifying redundant variables in models, assisting in their eliminatation to avoid an unlikely overfitting problem. In the case of the ANN model, we identified that the same four variables used in the LDA model were able to perfectly discriminate between enzymes and non-enzymes and assign each enzyme sequence to a specific subclass.

Finally, we also tested RBF models, which afforded results that were worse than the MLP models. In fact, the general accuracy was lower when compared to the MLP models, which usually need less neurons to achieve greater accuracy.

## 4. Materials and Methods 

### 4.1. Dataset

From the PDB, we retrieved a total of 81,486 protein FASTA sequences. Of those sequences, 26,073 were enzymes, while 55,413 were non-enzymes (α-proteins, β-proteins, membrane proteins, and so forth). Each of the 26,073 enzyme sequences belonged to one of the 65 enzyme subclasses. In order to avoid redundant sequences, we selected the enzymes using the specific EC classification query module of the PDB and then double-checked the dataset, eliminating double entries. Regarding the non-enzyme sequences, we randomly downloaded protein sequences belonging to different classes, such as membrane proteins, multi-domains, alfas and betas. The complete list of EC subclasses is reported in Appendix A, while Table 7 reports the number of entries for each one of the subclasses.

### 4.2. Molecular Descriptor Calculation

The software S2SNet [47] was used to transform each protein sequence into one sequence recurrence network (SRN). The SRN of a protein sequence can be constructed starting from one of two directions: (1) from a sequence graph with linear topology by adding amino acid recurrence information, or (2) from a protein representation graph with star graph (SG) topology by adding sequence information [48,49,50,51,52]. Note that, in both of these SRN representations of a protein sequence, the amino acids are the nodes and are paired (na and nb) in the network (being connected by a link, αab = 1) if they are adjacent and/or neighbour recurrent nodes. This means that αab = 1 if the topological distance between na and nb is d = 1 (chemically bonded amino acids), or if they are the nearest neighbour amino acid of the same type (A, R, N, D, C, Q, E, G, H, I, L, K, M, F, P, S, T, W, Y, V, X) with minimal topological distance, dab = min(dab), between them. The first node in the sequence (centre of the star graph) is a bias or a dummy non-residue vertex. 

Secondly, we needed to transform the SRN of each sequence into one stochastic matrix ^1^Π. The elements of ^1^Π were found by considering the probability (p_ab_) of reaching an amino acid (node nj) by walking from another amino acid (node ni) through a walk of length dij = 1 (Equation (2)):(2)$$p_{\mathrm{ab}}=\frac{\alpha_{\mathrm{ab}}}{\sum_{n=1}^{n=L}\alpha_{\mathrm{ab}}}\;$$

Note that the number of amino acids in the sequences was equal to the number of nodes (n) in the SRN graph, and was also equal to the number of rows and columns in ^1^Π, the length of the sequence (L), and the maximal topological distance in the sequence max(dab). In this work, we quantified the information content of a peptide using the Shannon entropy values (*θ_k_*) of the *k*-th natural powers of the Markov matrix ^1^Π. The same procedure was used to quantify the information of the q-seqs (*^q^θ_k_*) and r-seqs (*^r^θ_k_*). The formula for the Markov–Shannon entropy *^q^θ_k_* is as follows (Equation (3)): (3)θqkseq=−∑a=0a=Lpka⋅logpka  where pka represents the absolute probability of reaching a node moving throughout a walk of length *k* with respect to any node in the spectral graph. Further details of this formula can be seen in previous works [35,36,37].

In the Appendix A, we report the complete list of sequence entries with the respective value of the MD used to develop the models.

### 4.3. Multi-Target Linear model

The LDA model was developed using the General Discriminant tool implemented in the software STATISTICA [40]. The model is based on a multi-task approach, meaning it is able to predict if a sequence belongs to one out of the seven EC classes. It starts by identifying the presence of enzyme activity εq(ci) = 1 of subclass ci (or the absence of this activity εq(ci) = 0) for a query protein with a known amino acid sequence. The linear model is based on a linear equation, which directly correlates the dependent variable (enzyme or not) with the independent variable (MD). The multi-target LDA model was developed as follows. Once the MD were calculated, we computed the mean value of each subclass and then the difference between each sequence and the mean value of its subclass. Due to the model’s incorporation of the mean value of each subclass and the difference between each sequence, as well as the mean value of its subclass, the model is able to achieve a multi-target prediction. For further information regarding this statistical technique, please refer to the bibliography [53,54,55]. This same procedure was used also for the development of the multi-target ANN model. The validation of the model was performed using the cross-validation module implemented in the software. This procedure is aimed at assessing the predictive accuracy of a model. The test split the dataset into a training set and a validation set, ensuring that if an entry was included in the test set it could not be used in the validation test. In so doing, the model was developed using the cases in the training or learning sample, which, in our study, was 70% of the dataset. The predictive accuracy was then assessed using the remaining 30% of the dataset [56,57]. Standard statistics, such as the specificity (Sp), sensitivity (Sn), probability of error (p), cross-validation, and the Matthews correlation coefficient (MCC) [58], were used to assess the discriminatory power of the model.

### 4.4. Non-Linear Models

The non-linear models were developed using the neural network tool implemented in the software STATISTICA. In order to identify the best topology and architecture, we ran a large set of 50 models with various topologies. This step is crucial to avoid an (albeit unlikely) overfitting problem. We examined RBF and MLP networks since these usually perform better than other algorithms. The discriminatory power of the models was assessed using the cross-validation method. The models were validated using the cross-validation tool implemented in the ANN module of the STATISTICA software. In this validation procedure, the software automatically assigns 70% of the dataset to training the model. Once the model is trained, the remaining 30% of the inputs are used for validation. It is important to note that if an entry is used in the training set it cannot be used for the validation series.

## 5. Conclusions

Developing new, reliable, and robust methods for predicting protein function and enzyme class and subclasses is a key goal for theoreticians, especially in light of the recently introduced EC 7 class. In this work, we developed linear and non-linear models using an alignment-free approach to discriminate between enzymes and non-enzymes, as well as assign each enzyme sequence to a specific EC class. The best LDA model showed an overall accuracy of 98.63%, which is considered a remarkable result. However, we decided to explore further and develop some non-linear models using two different algorithms: MLP and RBF. While the latter was unable to improve the results of the LDA model, the MLP model was able to achieve an overall accuracy of 100%. This means that it was able to perfectly discriminate between enzymes and non-enzymes and identify the EC class of each enzyme.

## Figures and Tables

**Table 1 ijms-20-05389-t001:** Accuracy for the linear discriminant analysis (LDA) model.

	Training			Validation			Overall		
	All	−1 = Sn	1 = Sp	All	−1 = Sn	1 = Sp	All	−1 = Sn	1 = Sp
−1	98.13	40,781	778	98.27	13,613	240	98.16	54,394	1018
1	99.7	57	19,498	99.71	19	6498	99.71	76	25,996
Total	98.63	40,838	20,276	98.73	13,632	6738	98.66	54,470	27,014

**Table 2 ijms-20-05389-t002:** Relevant statistics for the LDA model.

Eigenvalue	CanonicalR	Wilk’sLambda	Chi-Sqr.	df	*p*-value	MCC
1.241879	0.744275	0.446054	49334.99	4.000000	0.00	0.97

**Table 3 ijms-20-05389-t003:** Performance of the best multi-layer perceptron (MLP) model found.

Obs. Sets ^a^	Stat. Param. ^a^	Pred. Stat. ^a^	Predicted sets		
			1	−1	nj
**Training Series**					
1	Sp ^a^	100	17,500	0	57,039
−1	Sn ^a^	100	0	39,539	0
total	Ac ^a^	100	17,500	39,539	57,039
**Validation Series**					
1	Sp ^a^	100	8572	0	24,445
−1	Sn ^a^	100	0	15,873	0
total	Ac ^a^	100	8572	15,873	24,445
**Overall**					
1	Sp ^a^	100	26,072	0	81,484
−1	Sn ^a^	100	0	55,412	0
total	Ac ^a^	100	26,072	55,412	81,484

^a^ Obs. Sets = Observed sets, Stat. Param. = Statistical parameter, Pred. Stat. =Predicted statistics, Sp = Specificity, Sn = Sensitivity, Ac =Accuracy.

**Table 4 ijms-20-05389-t004:** Resumé of the 10 best MLP and radial basis function (RBF) models.

		Training			Validation			Overall		
Model		−1 = Sn	1 = Sp	All	−1 = Sn	1 = Sp	All	−1 = Sn	1 = Sp	All
BEST MLP: 4-9-2	Total	55,412	26,072	81,484	55,412	26,072	81,484	55,412	26,072	81,484
	Correct	55,412	26,072	81,484	55,412	26,072	81,484	55,412	26,072	81,484
	Incorrect	0.00	0.00	0.00	0.00	0.00	0.00	0.00	0.00	0.00
	Correct (%)	100	100	100	100	100	100	100	100	100
	Incorrect (%)	0.00	0.00	0.00	0.00	0.00	0.00	0.00	0.00	0.00
1.MLP 4-7-2	Total	39,448	17,591	57,039	15,873	8572	24,445	55,412	26,072	81,484
	Correct	39,448	17,567	57,015	15,873	8562	24,435	55,412	26,034	81,446
	Incorrect	0	24	24	0	10	10	0	38	38
	Correct (%)	100	99.86	99.96	100.00	99.88	99.96	100.00	99.85	99.95
	Incorrect (%)	0	0.14	0.04	0.00	0.12	0.04	0.00	0.15	0.05
2.MLP 4-8-2	Total	39,448	17,591	57,039	15,873	8572	24,445	55,412	26,072	81,484
	Correct	39,448	17,565	57,013	15,873	8563	24,436	55,412	26,037	81,449
	Incorrect	0	26	26	0	9	9	0	35	35
	Correct (%)	100	99.85	99.95	100.00	99.90	99.96	100.00	99.87	99.96
	Incorrect (%)	0	0.15	0.05	0.00	0.10	0.04	0.00	0.13	0.04
3.MLP 4-10-2	Total	39,448	17,591	57,039	15,873	8572	24,445	55,412	26,072	81,484
	Correct	39,448	17,565	57,013	15,873	8563	24,436	55,412	26,037	81,449
	Incorrect	0	26	26	0	9	9	0	35	35
	Correct (%)	100	99.85	99.95	100.00	99.90	99.96	100.00	99.87	99.96
	Incorrect (%)	0	0.15	0.05	0.00	0.10	0.04	0.00	0.13	0.04
4.MLP 4-11-2	Total	39,448	17,591	57,039	15,873	8572	24,445	55,412	26,072	81,484
	Correct	39,448	17,566	57,014	15,873	8563	24,436	55,412	26,037	81,449
	Incorrect	0	25	25	0	9	9	0	35	35
	Correct (%)	100	99.86	99.96	100.00	99.90	99.96	100.00	99.87	99.96
	Incorrect (%)	0	0.14	0.04	0.00	0.10	0.04	0.00	0.13	0.04
5.MLP 4-16-2	Total	39,448	17,591	57,039	15,873	8572	24,445	55,321	26,163	81,484
	Correct	39,448	17,567	57,015	15,873	8572	24,445	55,321	26,139	81,460
	Incorrect	0	24	24	0	0	0	0	0	0
	Correct (%)	100	99.86	99.96	100.00	100.00	100.00	100.00	99.91	99.97
	Incorrect (%)	0	0.14	0.04	0.00	0.00	0.00	0.00	0.09	0.03
6.RBF 4-21-2	Total	39,539	17,500	57,039	15,873	8572	24,445	55,412	26,072	81,484
	Correct	39,520	16,426	55,946	15,855	8059	23,914	55,375	24,485	79,860
	Incorrect	19	1074	1093	18	513	531	37	1587	1624
	Correct (%)	99.95	93.86	98.08	99.89	94.02	97.83	99.93	93.91	98.01
	Incorrect (%)	0.05	6.14	1.92	0.11	5.98	2.17	0.07	6.09	1.99
7.RBF 4-29-2	Total	39,539	17,500	57,039	15,873	8572	24,445	55,412	26,072	81,484
	Correct	39,165	17,475	56,640	15,714	8561	24,275	54,879	26,036	80,915
	Incorrect	374	25	399	159	11	170	533	36	569
	Correct (%)	99.05	99.86	99.3	99.00	99.87	99.30	99.04	99.86	99.30
	Incorrect (%)	0.95	0.14	0.7	1.00	0.13	0.70	0.96	0.14	0.70
8.RBF 4-21-2	Total	39,539	17,500	57,039	15,873	8572	24,445	55,412	26,072	81,484
	Correct	39,526	16,138	55,664	15,868	7873	23,741	55,394	24,011	79,405
	Incorrect	13	1362	1375	5	699	704	18	2061	2079
	Correct (%)	99.97	92.22	97.59	99.97	91.85	97.12	99.97	92.09	97.45
	Incorrect (%)	0.03	7.78	2.41	0.03	8.15	2.88	0.03	7.91	2.55
9.RBF 4-28-2	Total	39,539	17,500	57,039	15,197	8571	23,768	53,008	26,060	81,484
	Correct	39,489	16,000	23,489	15,197	8448	23,645	53,008	25,674	78,682
	Incorrect	50	1500	1,450	0	123	123	0	386	386
	Correct (%)	99.87	91.43	95.65	100.00	98.56	99.48	100.00	98.52	99.51
	Incorrect (%)	0.03	7.78	4.35	0.00	1.44	0.52	0.00	1.48	0.49
10.RBF 4-26-2	Total	39,539	17,500	57,039	15,873	8572	24,445	55,412	26,072	81,484
	Correct	11,880	6629	18,509	4748	3170	7918	16,628	9799	26,427
	Incorrect	27659	10871	38530	11125	5402	16527	38784	16273	55057
	Correct (%)	30.05	37.88	32.45	29.91	36.98	32.39	30.01	37.58	32.43
	Incorrect (%)	69.95	62.12	67.55	70.09	63.02	67.61	69.99	62.42	67.57

**Table 5 ijms-20-05389-t005:** Quantitative analysis of the non-optimal MLP models.

Model	Class	Fail	Total Class
1. MLP 4-7-2	6.4	1	104
	6.5	34	36
2. MLP 4-8-2	1.6	3	4
	6.4	1	104
	6.5	34	36
3. MLP 4-10-2	1.6	3	4
	6.4	1	104
	6.5	33	36
4. MLP 4-11-2	1.6	3	4
	6.4	1	104
	6.5	32	36
5. MLP 4-16-2	6.4	1	104
	6.5	33	infer 36

**Table 6 ijms-20-05389-t006:** Sensitivity analysis for the artificial neural network (ANN) model.

Input Variable	Variable Sensitivity	Variable Name/Details
<Tr5(srn)>	15,896,991	Expected value of Trace of order 5 of the srn for the sequence
D Tr5(srn)	1,288,626	Deviation of Trace of order 5 of the srn with respect to the mean value of the class
<Tr3(srn)>	591,331.9	Expected value of Trace of order 3 of the srn for the sequence
D Tr3(srn)	108.7591	Deviation of Trace of order 3 of the srn with respect to the mean value of the class

**Table 7 ijms-20-05389-t007:** Number of entries for each subclass.

EC Subclass	Number of Sequences	EC Subclass	Number of Sequences	EC Subclass	Number of Sequences
1.1	555	2.3	722	4.6	120
1.2	250	2.4	424	4.99	95
1.3	172	2.5	291	5.1	176
1.4	108	2.6	19	5.2	74
1.5	5	2.7	3112	5.3	247
1.6	4	2.8	71	5.4	160
1.7	91	2.9	10	5.5	115
1.8	165	3.1	1559	5.6	159
1.9	73	3.11	7	5.99	3
1.10	555	3.13	3	6.1	277
1.11	136	3.2	700	6.2	38
1.12	32	3.3	164	6.3	291
1.13	123	3.4	1481	6.4	104
1.14	244	3.5	561	6.5	36
1.15	162	3.6	417	7.1	8827
1.16	173	3.7	69	7.2	927
1.17	121	3.8	77	7.4	189
1.18	45	3.9	3	7.5	187
1.20	250	4.1	486	7.6	197
1.21	28	4.2	460		
1.23	3	4.3	97		
2.1	522	4.4	39		
2.2	107	4.5	25

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
