# Peer review of "Alignment-Free Method to Predict Enzyme Classes and Subclasses"

_ijms, 2019, doi:10.3390/ijms20215389_

Round 1

Reviewer 1 Report

The reviewers chose not to directly address most of the issues raised in the first review round. Although they claim that in literature there is no model meant to deal with the latest EC labeling, this was not into question. The question was, as stated, whether these models could perform better also on the new labeling, and how the presented method would perform on the old one. The issue remains unaddressed.

The authors claim that making clear whether their performance is due to the novel training set or to their method is not in the aim of this paper. However, as stated, it is still not clear if they want to propose a novel method as more efficient than the previous ones or if they want just to show that the new EC labeling release is computationally predictable. In the former case, comparing with other methods would be necessary. In the latter, using a novel method would be misleading.

The required additional details about the validation procedure were added to the reviewers response but not to the manuscript.

Concerning the prediction errors, the author provided generic explanation, but the requested quantitative analysis is still missing.

Author Response

Dear reviewer,

please find attached the revised manuscript and responses to your issues.

Best regards

Reviewer 2 Report

This manuscript aims to present two computational methods, linear and non-linear, capable of predicting enzyme classes, including the new class recently created by the Enzyme Commission. The originality of the work stands precisely on the inclusion of this newly created class.

There are several questions in this manuscript that would probably be unclear for the general readers of the International Journal of Molecular Sciences, who are not necessarily familiar with some machine learning issues. In this sense, some points need further explanations.

The Enzyme Classification is a hierarchical classification that consists of four different levels. The models developed in this work classify enzymes in the first to levels of the hierarchy. If the dependent variable of the linear discriminant analysis shown in equation (1) is binary, how is the model able to classify enzymes into one of the 7 classes and 65 subclasses? Or is the linear model used only to discriminate between enzymes and non-enzymes? Similarly, with the aim to clarify the type of predictions obtained with the artificial neural networks, what is the codification of the output given by the two neurons in the output layer?

According to the results reported in this work, the best non-linear model was able to achieve an overall accuracy of 100%. Does this mean that the model correctly classified all enzymes into the 7 classes and the 65 subclasses? If the accuracy only refers to the enzyme/non-enzyme discrimination, the authors should also provide the performance of the predictive models in terms of the enzyme classes.

The dataset was divided into a training set containing 70% of the data, and a validation set containing 30% of the data. The authors state that the validation set was used for the validation of the models using a cross-validation procedure. It is not clear at all how this validation was performed, since in a typical cross-validation scheme there is a single dataset that is partitioned into two subsets, training and validation, this step being repeated several times and averaging the performance results obtained for the different validation subsets. The authors should explain in detail the validation methodology that they applied.

The performance of any predictive model is usually assessed in terms of the validation dataset. Are the results shown in Table 2 corresponding to the training set, the validation set or the overall results? If they do not correspond to the validation set, then these should be added.

Are the training and validation sets used for the development of the linear model different from the ones used for the development of the non-linear models? It seems to be so, according to the cardinalities of these subsets shown in Tables 1 and 3.

In Table 4:

- The results corresponding to the best artificial neural network, MLP 4-9-2, are missing.

- There is a repeated architecture: RBF 4-21-2.

- According to the cardinalities shown, it seems that the model RBF 4-28-2 was developed using datasets different from the other models shown in the same table.

As the authors admit, the mechanistic interpretation of artificial neural network models is far more complicated than the linear discriminant analysis model. It would be very helpful to add a paragraph explaining the meaning of the sensitivity parameter shown in Table 5.

The variables DTr3(srn) and DTr5(srn) represent some deviation “with respect to the mean value of the class”. What are these classes? It seems that there exists a kind of pre-classification. Could the authors explain further more on this issue?

Once the authors address all the points above, in my opinion the work could be considered for publication.

Author Response

This manuscript aims to present two computational methods, linear and non-linear, capable of predicting enzyme classes, including the new class recently created by the Enzyme Commission. The originality of the work stands precisely on the inclusion of this newly created class.

There are several questions in this manuscript that would probably be unclear for the general readers of the International Journal of Molecular Sciences, who are not necessarily familiar with some machine learning issues. In this sense, some points need further explanations.

The Enzyme Classification is a hierarchical classification that consists of four different levels. The models developed in this work classify enzymes in the first to levels of the hierarchy. If the dependent variable of the linear discriminant analysis shown in equation (1) is binary, how is the model able to classify enzymes into one of the 7 classes and 65 subclasses? Or is the linear model used only to discriminate between enzymes and non-enzymes? Similarly, with the aim to clarify the type of predictions obtained with the artificial neural networks, what is the codification of the output given by the two neurons in the output layer?

Dear reviewer,

As stated in the paper, in this work we reported a so-called multi-task or multi-target model. These models even if it seems they are a classical binary classification model, are based on a specific statistical theory which is able to classify more than a class in one of the two main classes. Regarding the output of the neural networks, it is the same as for the linear model.

According to the results reported in this work, the best non-linear model was able to achieve an overall accuracy of 100%. Does this mean that the model correctly classified all enzymes into the 7 classes and the 65 subclasses? If the accuracy only refers to the enzyme/non-enzyme discrimination, the authors should also provide the performance of the predictive models in terms of the enzyme classes.

Dear reviewer, the best ANN model is able to discriminate between enzymes and no-enzymes and, in addition, can classify all enzymes into the 7 classes and the 65 subclasses

The dataset was divided into a training set containing 70% of the data, and a validation set containing 30% of the data. The authors state that the validation set was used for the validation of the models using a cross-validation procedure. It is not clear at all how this validation was performed, since in a typical cross-validation scheme there is a single dataset that is partitioned into two subsets, training and validation, this step being repeated several times and averaging the performance results obtained for the different validation subsets. The authors should explain in detail the validation methodology that they applied.

Dear reviewer, the model was validated using the cross-validation method implemented in the Statistica software. In this validation procedure, the software automatically assigns the 70% of the dataset for the training of the model. Then, the remaining 30% of the inputs are used for the validation of the model.

The performance of any predictive model is usually assessed in terms of the validation dataset. Are the results shown in Table 2 corresponding to the training set, the validation set or the overall results? If they do not correspond to the validation set, then these should be added.

Dear reviewer, in the Table 2 we have reported some relevant statistics for the LDA model, while in the Table 3 are reported the results for the training set, validation set and overall. Finally, in the Table 4 we have also reported the results in the training set, validation set and overall for the other ANN models

Are the training and validation sets used for the development of the linear model different from the ones used for the development of the non-linear models? It seems to be so, according to the cardinalities of these subsets shown in Tables 1 and 3.

Dear reviewer, in the LDA model we have randomly assigned to each input to the training or the validation set. In the case of the ANN model, is the software that for each model randomly assign the input to the training or validation set.

In Table 4:

- The results corresponding to the best artificial neural network, MLP 4-9-2, are missing.

Dear reviewer, the results of the MLP 4-9-2 are reported in the Table 3

- There is a repeated architecture: RBF 4-21-2.

Dear reviewer, since the ANN models were automatically developed by the software, sometimes the architecture may be repeated. Even if the architecture is the same, the results may be different as in the case of the two RBF 4-21-2 reported.

- According to the cardinalities shown, it seems that the model RBF 4-28-2 was developed using datasets different from the other models shown in the same table.

Dear reviewer, the results of the model RBF 4-28-2 was erroneously reported and corrected.

As the authors admit, the mechanistic interpretation of artificial neural network models is far more complicated than the linear discriminant analysis model. It would be very helpful to add a paragraph explaining the meaning of the sensitivity parameter shown in Table 5.

Dear reviewer, following your kind suggestion we have added a paragraph explaining the meaning of the sensitivity parameters shown in Table 5.

The variables DTr3(srn) and DTr5(srn) represent some deviation “with respect to the mean value of the class”. What are these classes? It seems that there exists a kind of pre-classification. Could the authors explain further more on this issue?

Dear reviewer, the variables DTr3(srn) and DTr5(srn)represent the deviation or the difference between the input and the mean value of its subclass while <Tr3(srn)> and <Tr5(srn)> represent the mean value of the subclass. These variables are the keypoint of the multi-target model. In fact, due to the use of the mean and the difference between the mean value of a specific subclass we can develop this kind of model.

Once the authors address all the points above, in my opinion the work could be considered for publication.

Round 2

Reviewer 1 Report

The main missing point form the manuscript was a detailed description of the validation procedure to make the results reproducible. Such description has now been added, making the manuscript acceptable. I will drop further requests on which the authors do not agree with me.

For the final version, please:

Check percentages in the non-optimal MLP models. For example, in the training set of the MLLP 4-16-2 model, 24 errors are reported but the corresponding percentage is reported as 100% instead of ~99.86%, which also affect the "overall" columns. Check the new paragraph at page 8, lines 141-146: "theassessment", "valuesfound", "can drastically draw". Consider mentioning the quantitative analysis of the errors produced by the sub-optimal models as reported in the response to reviewers. It shows that such models tend to fail systematically on class 6.5, which has almost 0% accuracy, both when they have more and when they have less parameters than the optimal model. This means that in the EC classification, the 6.5 subclass is particularly problematic to distinguish from other classes (from the analysis we don't know which ones in particular) and that the largest contribution of the optimal model is to fix this specific problem. 

Author Response

Check percentages in the non-optimal MLP models. For example, in the training set of the MLLP 4-16-2 model, 24 errors are reported but the corresponding percentage is reported as 100% instead of ~99.86%, which also affect the "overall" columns. Check the new paragraph at page 8, lines 141-146: "theassessment", "valuesfound", "can drastically draw". Consider mentioning the quantitative analysis of the errors produced by the sub-optimal models as reported in the response to reviewers. It shows that such models tend to fail systematically on class 6.5, which has almost 0% accuracy, both when they have more and when they have less parameters than the optimal model. This means that in the EC classification, the 6.5 subclass is particularly problematic to distinguish from other classes (from the analysis we don't know which ones in particular) and that the largest contribution of the optimal model is to fix this specific problem. 

Dear reviewer,

We have revised the statistics shown in Table 4. We are sorry for that error which was due to an approximation of the decimals. We have also checked the new paragraph at page 8. Finally, we have also added a table (Table 5) reporting the quantitative analysis and commenting that the non-optimal models are failing to classify the 6.5 subclass.

Reviewer 2 Report

The authors have addressed all of the issues raised by this reviewer, with different degrees of clarity. In my opinion, the work can be now considered for publication.

The following suggestions would improve the clarity of the manuscript:

I think it is still unclear to the readers how the linear equation (1) can classify into 65 different subclasses. It should be clarified to which data set correspond the results in Table 2: training set or validation set. The assumed “relevancy” of these results depends largely on it. The results corresponding to the best artificial neural network, MLP 4-9-2, could be repeated in Table 4 for completeness of the table. The authors could add a short explanation about the variables DTr3(srn) and DTr5(srn), given the key importance that they have in the model.

Author Response

I think it is still unclear to the readers how the linear equation (1) can classify into 65 different subclasses. It should be clarified to which data set correspond the results in Table 2: training set or validation set. The assumed “relevancy” of these results depends largely on it. The results corresponding to the best artificial neural network, MLP 4-9-2, could be repeated in Table 4 for completeness of the table. The authors could add a short explanation about the variables DTr3(srn) and DTr5(srn), given the key importance that they have in the model.

Dear reviewer,

We have added few lines explaining the general multi-target LDA principle.

Regarding Table 2, as usual in these cases, the data refers to the overall model.

Following your suggestion, we have added the results of the MLP 4-9-2 to Table 4.

Finally, regarding the variables, we have added few lines in the manuscript to explain those. However, please consider that the software was not developed by our group and due to this we can only refer to the user manual to explain the meaning of the variables.

This manuscript is a resubmission of an earlier submission. The following is a list of the peer review reports and author responses from that submission.

Round 1

Reviewer 1 Report

This paper makes two flavors of model of EC number from a number of descriptors that use sequence information but do not require alignment. Not surprisingly, good cross-validation is seen.

I have a number of issues with this paper. I apologize if some of the needed information is in Supporting Material; I was not able to download it.

1.     I am not familiar with the descriptors discussed in this paper, and I did not find the explanation provided here very enlightening.  I cannot tell, for instance, if the descriptors use only sequence information or structural information also, although the fact that the data comes from the PDB implies structural. Did the authors try any simpler, more interpretable, descriptors using sequences alone, like residue pair or residue triplets, to see if these worked as well?

2.     Nothing is said about how proteins in the PDB are selected. The PDB is heavily biased, containing multiple entries of the same proteins (e.g. HIV protease, T4 lysozyme, etc.) This could make the cross-validation very optimistic because identical proteins would be in the training and test sets.

3.     It is not clear how the models were built. One could have one model per EC subclass and measure the probability of any one protein being in the one subclass, or you could build a multi-task model that tries to predict all subclasses simultaneously.

4.     Table 4 contains many types of model (MLP 4-8-2, RBF4-21-2, etc.) but I don’t know what these mean.

5.     The important descriptors in Table 5 are not interpretable to biologist.

6.     I cannot tell if feature selection is done before or during cross-validation. Doing it before is usually considered “cheating.”

7.     The discussion in the introduction exaggerates the importance of feature selection. Recursive partitioning methods (random forest, boosting, etc.) do not care if there are many non-relevant descriptors.

Reviewer 2 Report

The authors apply ML techniques to identify EC subclasses for to proteins based on their sequences after conversion to sequence recurrence networks.

One of the major claims of this study is the necessity of coping with the new EC class 7 introduced a few months ago. However, this does not justify the introduction of any additional method, as we don't know, and the paper doesn't demonstrate, whether previous approaches were able or not to correctly predict class 7 after simply relabelling the input data.

The paper claims 100% accuracy for the presented ML method, while even a linear model (LDA  after feature selection) achieves near-perfect accuracy. However, no comparison is made with performances of previous methods. If other methods don't achieve 100% accuracy, then the authors should highlight that their approach works better, with or without class 7. If the introduction of class 7 improves the results, than they should assess whether their method or a better EC labelling is responsible for the improvement. It is not totally clear to me if the authors are proposing a classification method or if they just want to show that EC class 7 can be correctly predicted with a standard approach.

Important details are lacking about the train / validation / test phase. In particular, the test set used to choose the best MLP should be different from any set used in optimizing each MLP separately. It is not clear if this was done.

Other details that should be reported include the distribution of the EC labels, as it is relevant to know at least what fraction of the used labels corresponds to the new EC 7. When an MLP model doesn't achieve 100%, which class is it failing with?

Some introduction of the used data should be moved before the description of the results, otherwise it is difficult to make sense of them.

The ROC curve of a 100% accuracy test does not carry any additional information as compared to the accuracy score itself and can be safely removed.